# Inclusive Culture in Compulsory Education Centers: Values, Participation and Teachers’ Perceptions

**DOI:** 10.3390/children9060813

**Published:** 2022-05-31

**Authors:** Rosa-Eva Valle-Flórez, Ana María de Caso Fuertes, Roberto Baelo, Rosario Marcos-Santiago

**Affiliations:** 1Department of General and Specific Didactics and Educational Theory, University of León, 24071 Leon, Spain; rosa-eva.valle@unileon.es; 2Department of Psychology, Sociology, and Philosophy, University of León, 24071 Leon, Spain; amcasf@unileon.es (A.M.d.C.F.); mrmars@unileon.es (R.M.-S.)

**Keywords:** inclusion, inclusive culture, special educational needs (SEN), inclusive education, teacher training, compulsory education

## Abstract

This article explores teachers’ perceptions concerning educational inclusion as part of an inclusive culture. The study focuses on compulsory education from the teachers’ point of view. We used three factors indicated in the “Index of Inclusion”: inclusive values, degree of participation in the educational community, and the teachers’ perceptions of the educational response offered to SEN students. To comply with the proposed objective, we explored nine variables to understand their influence on the attitudes of teachers and other professionals towards educational inclusion. These variables were gender, age, teaching seniority, educational stage, professional profile, type of center, geographic location of the center, years of experience and characteristics of SEN students, as well as the training received to meet the needs of all students. We found significant differences in the variables of age, educational stage, student characteristics, and training received, and recommendations are provided to address the needs detected.

## 1. Introduction

Culture can be understood as an integrated set of learned behavioral traits that are manifested and shared by the members of society. Its essential elements are the values, conceptions, and guidelines for actions within a community, which define what is *good*, *bad*, *desirable*, or *rejectable*. From them, a whole normative fabric is generated by which the conduct of its members is governed and serves as a reference to evaluate the behavior of others. Values and responses to people considered different or disabled have changed throughout history and cultures, ranging from religious demonization, medical pathologization, discrimination, and normalization to the defense of human diversity and the fight for social inclusion. For Castellanos and Lucero [1], “disability is defined within each culture according to the imaginary and the meanings that are held about the body.” For Ferreira [2], the same society that defines the (social) identity of the disabled is the one that culturally and symbolically “*disables*” the disabled. People with disabilities are perceived as “*the others*”, they are not within the norm, who “*are not normal*”, so they present a deviation regarding the expectations of behavior and values of a specific society in a particular moment. In this way, disabilities “*involve social processes of breaking norms, labeling and, often, dehumanization*” [3].

To understand the process of disability, it is essential to reflect on Goffman’s approach [4] from a paradigm of symbolic interactionism. He studied how stigmatizing labels, understood as social marks with negative connotations that society uses to define people, serve to show violations of the norm. He introduced the concept of social identity to describe the personal qualities that remain constant through situations. These identities are consolidated by the reactions of others towards us and how social judgments can create stigmatized identities that are difficult to avoid.

The categorization and tagging of students and the use of the terms “SEN-special educational needs” and “disability” in particular have awakened considerable confusion while highlighting the existence of different ideological approaches [5,6]. In recent decades, progress has been made towards an approach to disability linked to the social model of disability. This adoption has meant a significant change in the study and assessment of disability. This theoretical approach focuses on individuals and their insufficiencies in doing so on environments and social attitudes, focusing on how prejudice permeates cultural representations, language, and socialization itself. In this line, Barnes [7] pointed out that “*while social responses to insufficiency are not universal, there have been permanent cultural prejudices against people who suffer from insufficiencies*”, emphasizing the relationship between culture and the oppression suffered by disabled people.

Within this approach, an inclusive society must encompass all the people’s diversity, emphasizing acceptance and respect for diversity among the various human identities linked to issues such as race, gender, sexual orientation, religion, and also disability, avoiding previous judgments about them and their pathologization or labeling [8].

There is a consensus that educational institutions form the fundamental axes in the socialization process, together with families. Furthermore, through the Education for All initiative, UNESCO [9,10,11,12,13] has promoted forums and declarations on the need for inclusive and equitable quality education that promotes equal opportunities and lifelong learning for all. In this regard, Casanova [14] carried out a complete normative analysis that confirms the existence of a solid normative framework based on the principles of inclusive education, which indicates the presence of sufficient evidence [15,16,17] to affirm that there is a new paradigm that allows interpreting and intervening on human differences in the educational field. Furthermore, under this new paradigm, numerous actions and cases of good practices have been analyzed in the academic context in which inclusion is conceived as a process of continuous improvement that allows the identification and elimination of barriers or factors that limit the possibilities of participation and the educational success of all students.

However, inclusive education continues to be a challenge; it involves a change in culture and values since the focus must not be solely on students with specific needs but also on the changes and actions required by teachers, students, and their families. This aims to identify the barriers that underlie low performance and participation and representation within society [18].

This work is intended to investigate school culture, the values, beliefs, and interactions that concur in the compulsory education centers, understanding them as guides that favor or, on the contrary, hinder the behaviors, practices, and participation of the education community in achieving an inclusive culture [19].

## 2. Background

A review of the different meta-analyses works that have been developed on this subject [20,21,22,23] indicates that the most studied factor within the field of compulsory education and concerning inclusive culture is based on the relationships between the perceptions of teachers towards students with Special Educational Needs (SEN) and other study variables. This review also reveals the lack of studies that collect the voice of students at these educational levels (from 6 to 16 years old) and the fact that there are few that deal with inquiring about family opinions. In addition, practically all the studies have been based on Likert-type questionnaire application designs, taking teachers as a sample, specifically in their initial training phase (mainly with students in training). They collected opinions before professional practice and in the first stages of service (pre-service/in-service). They analyzed them with variables such as gender, age, teaching experience with students with special educational needs, the training obtained about the types of needs, and educational support measures. However, the conclusions of these studies are not conclusive. Lacruz-Pérez’s paper [24], which analyzes eighteen studies to look for the differences in the perception of attitudes towards inclusive education concerning gender, points out that seven of them show how women tend to be more positive towards inclusive education, both in the initial training and in the development of professional practice [25,26,27,28]. However, other works affirm the opposite, pointing out that men have better attitudes towards inclusive education [29,30]. This paper [24] also highlights results that indicate that teaching experience positively correlates with favorable attitudes towards educational inclusion [31,32,33]. It is also shown how other investigations define the association between the training that teachers have with the presence of a disposition, an attitude more favorable to respond to the specific needs of the students [34,35].

Similarly, in the research mentioned above, a relationship can be seen between the favorable predisposition toward educational inclusion and in which educational stage they are working. Thus, the most positive attitudes among teachers decrease as the level of the educational stage progresses. There is a more significant positive predisposition toward the educational inclusion of early childhood education teachers than those who work in primary education, who also have a more favorable inclination than those in secondary education. On the contrary, few studies concerning the variable “*contact with people with SEN*” according to the students’ degree or type of educational need, or other personal and environmental characteristics.

## 3. General Purpose

This research seeks to describe and analyze inclusive culture in compulsory education centers, taking the province of Leon (Spain) as the geographical framework of reference. For this, practicing teachers’ opinions and perceptions were investigated to identify the barriers that hinder inclusive education.

## 4. Materials and Methods

It was decided to develop work under a design non-experimental and descriptive, using mixed techniques to analyze the data obtained. A questionnaire developed and validated ad hoc for the study was used to obtain the quantitative data. Depending on the nature of the data obtained, the analysis was carried out using non-parametric techniques to associate variables. This study was complemented with contributions of qualitative data (from a discussion group) to deepen the understanding of quantitative data, determine the barriers that prevent or limit educational inclusion, and propose corrective measures according to the situations perceived as susceptible to improvement.

### 4.1. Participants

The regional statistical services (http://www.jcyl.es/sie/v2/educaCv2irAmodulo.html, accessed on 20 April 2022) reported the existence of 2583 teachers in compulsory education. This number formed our population (N = 2583). 67.56% (N = 1745) belong to kindergarten and primary education, and 32.44% (N = 838) to compulsory secondary education. The percentage of teachers in public education is 76.28%, and 23.72% in private-concerted education. The percentage of women is 64.58%, and 38.42% are men. A formula was used for finite populations to calculate the sample size, with a confidence level of 95% and a precision range of 2.5% [36], resulting in the need for 335 subjects to ensure the sample’s representativeness.

A stratified random sampling of centers was carried out considering the criteria of ownership of the center (public/concerted-private), location (rural or urban), and educational stage. Based on this selection of centers, and with the collaboration of the director, several volunteer participants were requested proportional to the population data in terms of sex and educational stage, with the condition that they had had students with specific educational support needs in their classes.

The selected teachers belonged to twenty-eight ordinary educational centers located in the province of León (Spain). In our country, all publicly funded schools (private and public schools) must provide schooling for students with educational needs, except for private schools without financial support from the state and specific special education centers that provide schooling for students with very severe SEN or multiple handicaps. These centers and their teachers were excluded because they were not the object of the study. The sample of this study was composed of 311 teachers and eleven teachers after 17 questionnaires were eliminated because they were incomplete or did not meet the indicated requirements. The sample comes from 28 educational centers located in the province of Leon (Spain). It was randomly selected among the 215 Early Childhood, Primary, Secondary, and High School education centers in the province mentioned above considering the criteria of ownership of the center (public/subsidized/private), location of the center (rural or urban), educational stage in which they teach, and on the condition that they have had students in their classes who need specific educational support. Therefore, the sample maintained the trend in proportion to the population data shown in the previous paragraph. The data was triangulated to generate a series of data that was as heterogeneous and representative as possible in terms of characteristics, location, and typology of the educational center.

The discussion group consisted of six professionals with extensive teaching experience: two Primary Education teachers and the other two from High School. Four were special education teachers, and the other two were not. Three were teachers at rural schools, while the other three were teachers in urban schools. It should be noted that the six participants were officials belonging to the public education system of the province of León (Spain). They had completed the questionnaire previously, considering the criteria followed to make up the questionnaire’s sample. They were five women and men of different ages (ranging from 34 to 59 years old). They were considered key informants and representative of the professional profiles involved in inclusion: tutors, subject teachers, management teams, and support professionals. It should be noted that the six participants were officials belonging to the public education system of the province of Leon (Spain).

### 4.2. Instruments and Procedure

The questionnaire was developed specifically for this study. It was designed for more comprehensive research led by the research group Diversity Disability and Special Educational Needs of the University of Almeria, which collected data from many other schools in different Spanish provinces. Authors validated it and found three significant factors by principal components analysis: organizational and curricular aspects, composed of nineteen items; faculty and resources, which included seventeen items; and inclusive culture, which was conformed of twenty-two items [37]. These factors had an explained variance of 62.617%. Cronbach’s alpha for the whole questionnaire was 0.84, which is satisfactory since it yields scores above 0.80.

We only used the third-factor items (Cronbach’s alpha = 0.90) to look for differences in inclusive culture across the eight independent variables considered. This factor was composed of nine items related to inclusive values, eight items related to participation in the educational community, and five items of teachers’ perceptions. All these items made up the dependent variables of our study.

As for the dependent variables, twenty-two items referring to inclusive culture were collected and grouped into three categories (Table 1). These categories were based on those described by Booth [17] and the indicators on inclusion prepared by the European Agency for the Development of Education for Students with SEN [34]. We defined each of them as:Inclusive values. Values guide and are the foundation of our actions. They are what give meaning and purpose to our efforts. Values affect the content of what we teach, how we teach, and how we relate to students and their families. In this sense, the items associated with this category aim to check the presence or absence of inclusive values in schools, such as equality, respect for differences, and non-discrimination. We asked the teachers if the criteria and indicators they used were consistent with inclusive principles to establish the center’s access, the formation of groups, the organization of support for SEN, and the methodology of curricular and complementary activities.Participation in the educational community. Teaching involves students, their families, and other professionals. This category gathers information about who usually participates in schools and the opportunities they must be involved and accepted to participate in decisions about their children’s school life.Perceptions. Referring to teachers and other SEN support professionals’ beliefs concerning the learning abilities of SEN students in mainstream schools. The impact of their schooling on their peers’ learnings, the training, and perceived self-efficacy to cope with the needs of students perceived as “different”.

The response format to the items was Likert-type with five response alternatives. The score ranged from 1 to 5, increasing the level of agreement, success, or frequency, depending on the formulation of the question.

To construct the factors and items under study and perform content validity, a bibliographic review of the subject was carried out, and the analysis of similar scales. The initial battery of items was submitted to a validity procedure through “Expert Judgment” among expert university professors in this field of study. To analyze the scale’s psychometric properties, a previous pilot study was carried out with samples of teachers from different autonomous communities of the state. First, principal components analysis was performed, identifying three factors with an explained variance of 62.617%, and the Kaiser-Meyer-Olkin sample adequacy measure (KMO = 0.922), and Bartlett’s sphericity test (<0.000) to determine the factorial adequacy of the instrument, resulting in suitable in the different factors analyzed. Next, the internal consistency of the questionnaire was analyzed using Cronbach’s alpha (α) and McDonald’s omega (ω). As shown in Table 2, the scale’s reliability was satisfactory since it yielded scores above 0.80.

Once the bibliographic review, design, and validation of the instrument had been carried out, the centers and teachers were randomly selected according to the criteria determined in the independent variables, the willingness to collaborate freely, and based on the census provided by the Provincial Directorate of Education of Leon (Spain).

The centers were contacted by telephone, and the project and the research objectives were presented to them. They were informed about the anonymity and confidentiality clauses on data collection. After this first contact, the questionnaire was sent both online and on paper by regular mail, depending on the preference of the center’s management, and the reminders and deadlines for completion or collection of the questionnaires were established. We must highlight and thank the high level of collaboration of the teaching staff and the management teams. Subsequently, the data were coded and analyzed to arrive at provisional quantitative results complemented by the qualitative data obtained by the discussion group. The composition of these groups was under the professional profiles and circumstances indicated in the independent variables: tutor teachers, specialist educational support personnel, and members of management teams with heterogeneous professional experience and the center’s location to generate a richer discourse.

## 5. Results

### 5.1. Sample

Our study sample was finally made up of 74.6% female teachers. The highest percentage of the sample (37%) was in the age range of 45 to 54 years, and a similar percentage (36.7%) had professional experience of more than 20 years. Of the sample, 67.5% worked in public schools (Table 3). Over half of the sample (51.4%) had between 5 and 20 years of experience working with SEN students. The sample had a similar percentage of teachers who worked in localities representing different population densities. Different professional profiles included the main specializations of the teaching staff grouped into the two selected categories: professionals specialized in supporting inclusion and professionals from different academic disciplines. Therefore, we can indicate, based on the population data of the registered categories, that the sample represented the proportional tendency of the different professional roles and was representative about sex, age, teaching experience, location, and type of center education of teachers in the province of Leon.

The Kolmogorov-Smirnov normal distribution test with Lilliefors correction was performed to apply the appropriate analyses, finding a significance of less than 0.05 in all the variables studied, so the null hypothesis of normality was rejected and non-parametric analyses were used.

### 5.2. Descriptive Analysis by Category

We present the results by the categories indicated in the structure section of the questionnaire to collect the most relevant contributions by the dimensions under study: inclusive values, participation in the educational community, and perceptions.

The opinion of teachers about the existence of “*inclusive values*” in schools was highly positive. Of the teachers, 80.71% considered that “almost always” or “always” (Figure 1) individual differences are considered, and academic achievement is related to the efforts and barriers that must be overcome.

In the category referring to the “*degree of participation in the educational community*”, 71.06% of the teaching staff considered that almost always or always there is collaboration and coordination necessary to attend to the needs of the SEN student between the teaching staff and professionals who intervene in the educational process (Figure 2). In addition, flexibility and opportunity are provided for families to engage and participate in their child’s learning.

Regarding the category related to “*teachers’ perceptions of the educational response that should be offered to SEN students*”, the data provided contradictions with the results obtained as an average in the values dimension (Figure 3). This circumstance could indicate that individual differences are not perceived as a factor of enrichment, expressing concern about a possible decrease in the academic level of the rest of the students due to the classroom heterogeneity. Similarly, this result shows that the concept of therapeutic intervention is still present, with resistance to assuming the commitment of the specific needs of the students.

### 5.3. Quantitative Results of Our Study According to Different Variables

To apply the appropriate analyzes, first, the Kolmogorov-Smirnov test with Lilliefors correction was performed to determine whether the data were normally distributed (null hypothesis of the analysis). When finding a significance of less than 0.05 in all the variables studied, the null hypothesis of normality was rejected, and non-parametric analyzes were chosen. The Mann-Whitney U for the study of two independent groups (gender, educational stage, professional profile, and center by funding/ownership), and the Kruskal-Wallis H for the study of three or more independent groups (age, teaching experience, center by the number of inhabitants in which is located, and teaching experience with SEN students) were used.

As the data did not follow a normal distribution, we did not work with means, but rather ranges were used to contrast the hypothesis that K samples were obtained from the same population. This is a matter of contrasting whether different independent samples are equally distributed and that, therefore, they belong to the same population, and comparing medians can be considered.

#### 5.3.1. Results by Gender

A Mann-Whitney U analysis was performed to check whether there were differences between men and women, which compares the results of two independent groups. No statistically significant differences were observed between male and female teachers to the answers provided in the questionnaire. A statistically significant difference (*p* = 0.022) was only observed between men and women when asked whether only support specialists should treat SEN students, with women mostly supporting this statement.

The magnitude of the effect size was calculated using the Hodges-Lehmann estimate, which uses the median differences between the two groups through the biserial rank correlation (r_b_), interpreting an effect size as irrelevant if it is <0.1; small (0.1); medium (0.3) and large 0.5 [38]. In this case, we found in item Cpe.2 a statistically significant difference with an association strength close to a medium-size value (r_b_ = 0.22) between the female gender and the perception that support for specific needs of students is a task for support specialists. However, in the discussion group analyzed, no differences were observed in the arguments used due to gender.

#### 5.3.2. Results by Age

To compare those aspects with three or more independent groups, non-parametric analyzes of the Kruskal-Wallis test were used (Table 4). In this way, the age of the participants was divided into four ranges: those between 24 and 34 years old, those between 35 and 44 years old, those between 45 and 54 years old, and, finally, those 55 years or older.

The analyzes showed significant differences in the different age groups of the teachers regarding the items of the category “*Participation in the educational community*” above all. In addition to item CV.6: The academic achievement of the SEN students is related to difficulties and barriers that must be overcome” from the “*Inclusive Values*” category.

After performing post hoc analyzes (Table 5), it was observed that these statistically significant differences occurred mainly among the group of younger teachers (between 24 and 34 years old) to the rest of the age groups. In this way, it can be said stated that the youngest teachers are the most “optimistic” regarding the degree of participation of the “Educational Community in the attention to students with Special Educational Needs”, and those who most firmly believe that there is a relationship between the effort invested and the barriers overcome by the SEN students and their achievements. Furthermore, a significant difference was also found between teachers who are between 35 and 44 years old and those 55 or older in the item that stated that in the centers, you keep families informed of the progress of their SEN children associated with disability, the older ones being those that affirmed this fact more forcefully.

When using categorical variables to calculate the strength of the effect size, Cramer’s V was used. Its interpretation indicates if the effect size is irrelevant if <0.1; small if 0.1; moderate (0.3); and large 0.5. The results obtained in the association of the significant variables indicated that the size of the effect was between small and moderate in the items indicated below: CV.6 (V = 0.15); Cpa.3. (V = 0.2); Chp.4. (V = 0.14); Chp.6. (V = 0.16); Chap.7. (V = 0.12); Chp.8. (V = 0.2).

#### 5.3.3. Results by Teaching Seniority

When the teaching seniority of the sample was compared, we obtained only three variables whose differences between the four groups (less than 5 years of teaching, between 5 and 10 years of teaching, between 11 and 20 years of teaching, or more than 20 years of teaching) were statistically significant: one item regarding the Participation in the educational community and two related to the teachers’ perceptions about the degree of how SEN students are perceived to be included in the Center (Table 6).

After the post hoc analyses were carried out, it could be verified how the statistically significant differences occurred between the group with less than 5 years of teaching experience and the rest of the groups, those with less teaching experience obtaining the highest scores. There were also statistically significant differences between those with between 5 and 10 years of teaching experience and those with more teaching experience, also in favor of those with less teaching experience. Therefore, there is a relationship between the teacher’s experience and the perception of an inclusive culture. The less the teaching experience, the more favorable the perception. However, it should be noted that there were not many indicators in which statistically significant differences were found (Table 7).

The strength of the effect size performed by Cramer’s V demonstrated that the strength of association between the items and the referenced age groups was small: Cpa.3. Teachers feel empowered to facilitate the participation of families (V = 0.15); Cpe.2. Only the support specialists should attend to SEN students since they are the ones trained for it (V = 0.12); Cpe.4. All students can achieve the general objectives in an inclusive classroom with the necessary help (V = 0.13).

#### 5.3.4. Results by Educational Stage

Concerning the educational stage at which the surveyed teachers teach, we observed statistically significant differences in the three categories evaluated: “*inclusive values*“, “*degree of participation in the educational community*”, and “*teachers’ perceptions of the educational response that should be offered to SEN students*”. In this way, both the inclusive values and the degree of participation in the educational community were higher in the Early Childhood and Primary Education stage than in the Compulsory Secondary Education and Baccalaureate stage. At the same time, the perceptions were more favorable in this latter educational stage (Table 8).

In this case, items CV.3: The work done by all students (with and without disabilities) is equally valued; Cpa.3: Teachers feel empowered to facilitate the participation of families; Cpa.4: There is flexibility in the class groups that encourages collaboration and communication between students; Cpa.5: The support provided by the Integration Support Classroom is coordinated with the work in the ordinary classroom; Cpe.5: For attending SEN students, it is necessary to start from what they know how to do and not from the general level of the classroom and Cpe.1: Teachers think that classrooms that have Sen students have lower academic levels than those that do not have these students, showed statistically significant differences with a strength of association between small and medium (r_b_ = 0.21). The effect size was smaller (r_b_ = 0.12) in items CV.4: SEN students actively participate in all activities center (exhibitions, works, musical activities, theatrical performances); Cpa.6: The Center offers families various ways to motivate them to get involved in their children’s learning and Cpe.2: Only the support specialists should attend to SEN students since they are the ones trained for it.

When we analyzed the discussion group, where four participants teach in the Early Childhood Education and Primary Education stages and two who teach in the Compulsory Secondary Education and Baccalaureate stages, the greatest difference observed revolved around who should promote the change of attitude of the students when facing “the future”. In this way, the Compulsory Secondary Education and Baccalaureate teachers believed that it is a work for the educational center.


*“let them leave the center with a changed mind, so when they have children, they would understand that education is the priority”*
(1:06:55)

Faced with this statement, a Primary Education teacher stated:


*“and do you think we can do that from the educational center?”*
(1:07:12)


*“it is very difficult to fight for something that you live in your house”*
(1:07:30).

These differences can be explained by the fact that in Compulsory Secondary Education and Baccalaureate, the students are older and are closer to emancipation and that the family has ceased to be the main source of influence on them. However, in Primary Education, between the ages of 6 and 12, children still depend on parents who have a predominant role in their development.

#### 5.3.5. Results by Type of Center

The information collected according to the type of center where the surveyed teaching staff teach showed multiple significant differences in the three categories evaluated, especially in terms of “*degree of participation in the educational community*” and “*inclusive values*”. In the category of “*teachers’ perceptions of the educational response that should be offered to SEN students*”, only in the item “All students can achieve the general objectives in an inclusive classroom with the necessary help” were significant differences found between teachers from public schools and teachers at subsidized schools (Table 9). In all cases, the differences indicate a greater awareness of “*inclusive values*”, a greater “*degree of participation in the educational community*”, and more favorable “*teachers’ perceptions of the educational response that should be offered to SEN students*” among teachers who work in subsidized education centers than those who work in public school.

We found, in all items, statistically significant differences with a strength of association between small and medium: The items with the smallest effect size were CV.1 and Cpa.5 (r_b_ = 0.1), followed by items CV.5; CV.7; CV.8; Cap.3 Cap.5, Cap.6; Cpa.7 (r_b_ = 0.2). The items with a larger effect size (r_b_ = 0.3) were CV.2; CV.6: Cap.1; Cap.2; Cap.4; Cpe.4. The median value of the three categories resulted in an effect size of 0.2 using the Rank-Biserial Correlation.

Since the participants of the discussion group all belonged to the public school, we cannot corroborate these results obtained at a qualitative level, but a biased vision of the public-school teachers was observed regarding the inclusive values and showing their discontent with issues such as


*“…lack of educational response adapted to these kids”*
(10:12),


*“…lack of teacher training to accept diversity”*
(9:10),

or the


*“…curricular adaptations are not valid”*
(39:10)

as well as the lack of resources such as human, technical, and time, which prevent the application of true inclusive policies:


*“…we have few professionals,” “we are patching up, very little by little”*
(12:34)


*“…We have to educate that way; it is much more difficult to accommodate a different person”*
(1:34:00).

This negative vision extended to the participation in the educational community, which was shown through expressions such as:


*“teachers who call themselves wonderful experts… but they are not”*
(18:49)


*“there is an attitude of I don’t say anything because if not… there is a mess of papers… let’s see who calls the mother and tells her…”*
(29:03)


*“…I will mark the student with a 5 because otherwise, I will have problems”*
(39:40)

and they demand greater coordination between teachers since they stated that there was still resistance to “*Special Education*” teachers entering the classrooms (8:47; 32:18).

In Secondary Education, they also pointed out the lack of stability and continuity in the staff of public education centers, which contributes to less involvement.


*“…when the teachers begin to realize the reality of the school, well, a year or two years have passed, and they have to leave”*
(38:38).

However, there was a very positive assessment about the participation of the students’ families, and they recognized the existence of intense work with the students’ parents:


*“…coordination does not have to be only between teachers, but also with families”*
(56:20)


*“…professionals have to take care of our students’ families”*
(56:35)


*“Parents value us a lot, and that is important for your personal self-esteem”*
(57:40).

#### 5.3.6. Results by Professional Profile

About the professional specialization profile of the teacher, we found that there were differences in the categories “*inclusive values* “and “*degree of participation in the educational community*” among the surveyed teachers who are specialists in supporting special educational needs (Special Education, Hearing and Speech and Educational Counselors), and the surveyed teachers who have a more general profile or belong to specialties other than attention to diversity (Table 10). In this sense, the opinion of the special education specialists was more negative than the rest of the teachers, having a lower score in the five items of “*inclusive values*” and the four items of “*degree of participation in the educational community*”. This circumstance could be attributed to the “*sense of loneliness*” that these professionals experience as they feel little or no support from the rest of the center’s educational team teachers.

In this case, the items with a medium strength of association and that could be considered to have a significant effect size (r_b_ = 0.3) were the items Cpa.2, Cpa.3, and Cpa.4 in the category on the “*teachers’ perceptions of the educational response that should be offered to SEN students*” that the teaching staff has about the participation of SEN students in the center life. The items that refer to the category of “*inclusive values*” (CV.1, CV.5, CV.6, CV.7, and CV.8) obtained a slightly lower effect size (r_b_ = 0.2).

This last perception was corroborated when we analyzed the discussion group since the specialists in support of special educational needs (two Special Education teachers, a Hearing and speech teacher, and an Educational Counselor from a Secondary Education Center) stated they were completely overwhelmed (4:11; 27:50) while calling for a truly inclusive educational model in which the weight of attention to diversity does not fall only on special educational needs support specialists


*“as much as we talk about inclusion… it is not done”*
(1:34:35)


*“no division of functions, that is, if we are 3 people working in the classroom,… it is not only me, it is not just the Special Education teacher… the three of us evaluate, the three of us teach… no division of functions… that is an inclusive classroom ”*
(1:17:24).

#### 5.3.7. Results by Years of Experience with SEN Students

When specifying the teaching experience of the participants in terms of SEN students’ attention, we found only two variables where statistically significant differences appeared: the total of the “*inclusive values*” (*p* = 0.029); and the item related to promotion, on the teachers’ part, of the collaboration and acceptance of all the students (Cpa.2) (*p* = 0.024).

In both variables, as the post hoc analyses showed, it was once again the teachers with fewer years of experience in this area who obtained the highest scores. In descending order, the most favorable age groups were 5 vs. 5–10 years, 5 vs. >20, and 11–20 vs. >20 years. In all cases, the strength of the effect size was small (V = 0.16)

#### 5.3.8. Results by the Number of Inhabitants of the Locality Where the Center Is Located

There were hardly any statistically significant differences between the groups depending on the number of inhabitants of the locality where the educational center of the respondents was located. Statistically, significant differences were only observed in the two variables. The first was the variable of “*degree of participation in the educational community*”, which states that “*the center offers families various ways to motivate them to get involved in the learning of their sons and daughters*” (*p* = 0.034). When doing post hoc analyses, we saw how the difference (*p* = 0.003) lay between the teachers who carry out their work in centers of localities with between 10,001 and 100,000 inhabitants (range = 65.13) and teachers who carry out their work in centers of localities with more than 100,000 inhabitants (rank = 86.06). This difference was positive in favor of the latter. The second was the “*teachers’ perceptions of the educational response that should be offered to SEN students*” variable that considers that only specialists in support of special educational needs should care for students with special educational needs since they are the ones who are fully trained to do so (*p* = 0.02).

Post hoc analyses found statistically significant differences between teachers in localities with fewer than 3000 inhabitants and teachers in localities between 3000 and 10,000 inhabitants (*p* = 0.002), teachers from localities with less than 3000 inhabitants, and those from localities with more than 100,000 inhabitants (*p* = 0.001), teachers from localities with between 10,001 and 100,000 inhabitants, and those from localities with more than 100,000 inhabitants (*p* = 0.025). In all cases, teachers from the most populated localities were the ones who obtained the best scores.

The strength of the effect size for the two items with statistical significance was Cpa.6: The Center offers families various ways to motivate them to get involved in their children’s learning (V = 0.14) and Cpe.2: Only the support specialists should attend to SEN students since they are the ones trained for it (V = 0.16).

## 6. Discussion and Conclusions

The results show a gap in inclusive education between the legal provisions and the practices developed in the educational centers. Many barriers continue to hinder the development of an inclusive educational model. In this sense, the issues that have been mentioned in relation to the lack of resources (human, time), support from the administration, and others are still relevant.

The perception of the professionals working in the educational centers about the inclusive culture is one of the best references to understand the depth of the assimilation of inclusive culture in educational centers [20,21,22,23]. This is because they are the last responsible for carrying it out, and its assimilation directly relates to the level of learning students achieve.

This research delved into the study of inclusive culture in compulsory education from the point of view of its teachers. We did it using the three factors proposed in the “Index of Inclusion” [17]: inclusive values, degree of participation in the educational community, and the teachers’ perceptions of the educational response offered to SEN students.

The quantitative data offered us an overview of the existing reality in the educational centers. This panoramic vision was enlarged and enriched by qualitative data analysis. In this sense, the combination of methodologies improved the work and provided a better foundation for the results. Thus, it was possible to delve into the discourse of teachers and investigate the perceptions and visions they have about real inclusion in schools and the barriers that hinder the development of an inclusive educational model that, at least initially, all regarded as ideal.

The questionnaire results regarding the teachers’ perceptions of “inclusive values” were highly positive. For example, between 75.5% and 86.1% of teachers considered that “almost always” or “always” the center’s performance indicators were in line with inclusive values; the difference was regarded as an enriching factor, and the design of activities respected individual differences according to the contributions of the Universal Design for Learning (UDL).

Several studies [38,39,40] have reported a significant association between SEN students and bullying at the school. Our study highlights that there is not a clear, direct relation between SEN and victimization; 38.9% of the teachers considered that SEN students might be bullied to a greater degree due to their characteristics than their peers, but 28% believe that victimization does not have to be associated with SEN, and 33.1% did not show a specific perception about it (Mdn: 80.7).

Concerning the degree of participation in the educational community, the average values of the items decrease (Mdn, 69.1). However, most of the teachers’ perception (87.8%) is that families are informed of their children’s progress with SEN (Cpa.8). On the other hand, when we asked if teachers feel empowered to facilitate the participation of families (Cpa.3), the results indicate the need for specific training, which reinforces the skills and strategies of teachers. Only slightly more than half (57.2%) felt they were trained, and the majority did not know how to deal with these situations. This request for training is a common demand among the professionals in the education, and not only in the compulsory stage [41].

The last factor, the teachers’ perceptions of the educational response offered to SEN students, represented the median least valued in the items of all the factors studied (Mdn, 31.2). The results lead us to conclude that the teachers’ perceptions of how, when, and by whom the educational intervention should be carried out are far from close to a desired inclusive culture. Thus, the measures of attention to diversity in the classroom and the teachers’ attitudes operate under an “integrative” vision associated with actions that discriminate against part of the student body, and that implies a dualization of education without being fully in line with the model of inclusive education. Again, these results agree with those found in other studies [16,23,42,43].

To comply with the proposed goal, we explored eight variables that can influence the attitudes of teachers and other professionals towards educational inclusion: sex, age, teaching experience, academic stage, professional profile, type of center, geographic location of the center, and years of seniority with SEN students.

Regarding gender, except for one item, no significant differences were found, contrary to the review by Lacruz-Pérez [20,23,24,44], who found most studies in which women were more proactive toward inclusive education in both initial and continuing education, women were also able to use more inclusive methodologies in teaching-learning processes.

In the age variable, we conclude that as the age of teachers increases, attitudes become more negative, as another study denotes [44,45]. Therefore, it would be appropriate to analyze the aspects that cause this change in subsequent studies.

The literature reviewed shows that the teaching experience positively correlates with the favorable attitudes toward educational inclusion [29,30,31]. As in previous studies [46,47], we found that the less teaching experience, which usually is associated with younger ages, the higher the rating of positive perception about the inclusive culture in the centers. However, the strength of the effect size was small.

Regarding the educational stage, as previous studies reflect [45,47,48], in the three categories evaluated in all the items, the score was higher in Early Childhood Education, followed by Primary Education, and worse than in Compulsory Secondary Education and Baccalaureate. Nevertheless, the strength of the association found was between small and medium between the educational stage variable and the items analyzed. These differences could be related to more significant variability in students’ abilities at the secondary school stage due to a broader curriculum gap and a more considerable workload for the teacher, which plays an important role when teachers evaluate inclusion [49]. However, we found more pessimistic opinions in professionals with expertise in educational support for SEN in the professional profile.

The same happens with the results related to the type of centers; public, subsidized, or private. There are no comparative studies for this purpose. We found multiple significant differences in the three categories evaluated, especially in terms of participation in the educational community and inclusive values in favor of the subsidized schools. The median value of the three categories had an effect size of 0.2. However, we must be very cautious when interpreting the results from this variable. The data on the distribution of the school population with SEN in our province, and Spanish schools in general, are not homogeneous in proportion or degree, or type of need between the network of public and (private-)subsidized schools. Subsidized and private schools have a notably lower proportion of students with SEN, with fewer learning difficulties, and their students come from a social stratum with more economic and educational resources. Moreover, other variables, such as the social and economic context or the level of education of the families, or the time available to dedicate to their children, can have an effect as variables that are not very favorable for learning and therefore be modulating variables that affect the type of center where the SEN students are schooled.

We did not find any differences worth mentioning related to the center’s geographic location or the greater or lesser size of the population-area where it is located, nor did we find studies to compare with our data. Regarding differences between countries, comparative studies have shown areas of development of inclusive education with different speeds and impacts. Anglo-Saxon countries such as Great Britain and Canada have been pioneers in implementing these policies. Therefore, it is reasonable to think that the more positive attitudes of teachers in these geographical areas are due to the experience and change in teachers’ attitudes, among other factors. However, we cannot forget that the paradigm of inclusive education is strongly linked with ideas of democracy, social justice, and citizen participation related to the countries’ policies and socio-cultural climate. These issues, in turn, require an investment in material and human support resources financed by the state to guarantee access to quality education in all social strata. A climate of civic democracy is a critical factor in developing inclusive education and conceptualizes how historical contexts affect attitudes and social values toward difference and disability and describes the structural and cultural barriers that occur in these countries for meaningful implementation of inclusive education [50].

Referring to years of experience with SEN students, we found some items associated with this variable, but the effect size was irrelevant. For example, in the literature reviewed, negative attitudes were more frequent in students with severe emotional or behavioral problems, cognitive problems, or great diversity in cultural, linguistic, and ethnic characteristics [49,51,52,53,54].

The qualitative data provided by the focus groups provide relevant information that supports the differences perceived by teachers depending on the origin and severity of the educational need. Other studies also found that attitudes were strongly influenced by the nature and severity of the disability and less by variables related to the teacher [55] and were in line with those of previous international reviews [20,21].

They also report a new variable related to specific practical information on how to intervene with SEN: the less training, the lower the perceived self-efficacy and the worse the attitudes [20,32,47,48,54,56]. A comparative study between teachers in South Africa and Finland, and its implications for teacher training, showed a higher perception of self-efficacy, particularly effectiveness in collaboration and behavior management, in professionals with more positive attitudes towards inclusion. Within the Anglo-Saxon context, the association between training and a better disposition to respond to the specific needs of students was shown due to a greater capacity to use inclusive methodologies in their processes [32]. This association between more positive attitudes and self-efficacy in inclusive methods is consistent with findings within the European context [56,57] and in other countries and territories [58,59]. The training variable affects the stages of compulsory education and university education [41], so recommendations to address these needs would reinforce these aspects.

### 6.1. Limitations

Although widely used in social science contexts, the use of opinion questionnaires contains the possible bias of the responses given due to “social desirability” that does not correspond to the authentic convictions or practices of the respondents. Therefore, it would be interesting to use a more significant number of critical informants through qualitative techniques, focus groups, in-depth interviews, or natural systematic observations in classrooms and centers to triangulate the data with other instruments and participants.

The analysis of our data is descriptive, not inferential. The use of multivariate data analyses such as logistic regression or structural equation methods would undoubtedly help select the most significant variables to make predictions. However, the purpose of educational research is transformation, i.e., the change of teaching practices rather than their prediction. Our study identifies the main barriers that hinder an inclusive culture, makes explicit routines and behaviors on which it is necessary to reflect, and can mark the roadmap. This roadmap demands changes from a focus based on teaching content to a student learning model focused on the competencies required in the society of the 21st century.

### 6.2. Improvement Proposals and Prospective

Finally, according to the data obtained and the conclusions of our study, and e review of the scientific literature on the subject, we believe that the Faculties of Education, in which initial training is developed, and the Centers of Ongoing Formation, must continue with the effort initiated to raise awareness and respect for human differences. Although it is true that progress has been perceived in all the factors identified in inclusive culture since study reviews carried out at the end of the last century, it is necessary to continue intensifying and deepening these values. There is a need for training in accessible curriculum design with proposals and concrete implementation of multi-level integrated projects and a Multi-Tiered System of Support (MTSS) [60,61,62,63].

This work opens new research lines. Unfortunately, there is a lack of studies concerning educational community members such as families, SEN students, and associations, so it is necessary to continue collecting the voices of all educational community members so that the research is complete. An increase in studies would permit exploring teachers’ attitudes, other support professionals, families, and the students themselves.

Moreover, it is necessary to undertake studies using experimental designs and inferential analyses that complement those merely descriptive and correlational. To this end, it will be necessary to expand data collection techniques with more reliable instruments such as in-depth interviews, observation records, or focus groups, which minimize the social desirability bias of questionnaires. Finally, it would be desirable for future studies to contemplate or make as explicit as possible the independent variables indicated in the conclusions to carry out comparative studies.

## Figures and Tables

**Figure 1 children-09-00813-f001:**
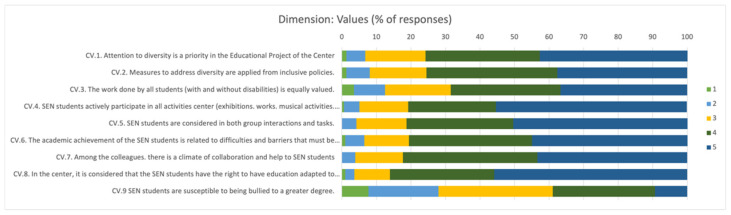
Percentage of responses about teachers’ opinions about the presence of inclusive values in their schools.1: Never; 2: Rarely; 3: Occasionally; 4: Almost always; 5: Always.

**Figure 2 children-09-00813-f002:**
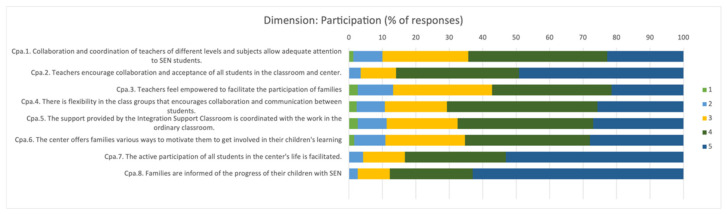
Percentage of responses about teachers’ opinions on the degree of participation in the educational community in the educational process. 1: Never; 2: Rarely; 3: Occasionally; 4: Almost always; 5: Always.

**Figure 3 children-09-00813-f003:**
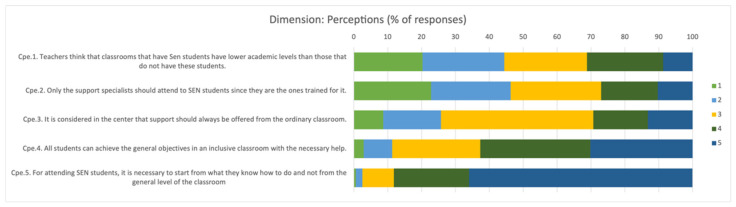
Percentage of responses about teachers’ perceptions of the educational response that should be offered to SEN students. 1: Strongly Disagree; 2: Disagree; 3: Undecided; 4: Agree; 5: Strongly Agree.

**Table 1 children-09-00813-t001:** Indicators of the questionnaire on inclusive culture.

Category	No. Items	Category Code
Inclusive values	9	CV.1–CV.9
Participation in the educational community	8	Cpa.1–Cpa.8
Perceptions	5	Cpe.1–Cpe.5

**Table 2 children-09-00813-t002:** Scale reliability statistics.

95.0% Confidence Interval				
	McDonald’ ω	Cronbach’s α	Lower	Upper
Scale	0.86	0.84	0.830	0.885
Values	0.90	0.90		
Participation	0.90	0.90		
Perception	0.70	0.65		

Note. Of the observations, 311 were used, 0 were excluded, and 311 were provided.

**Table 3 children-09-00813-t003:** Sample distribution. Categorization of the socio-professional variables of the sample.

		Fr	%
Sex	Women	232	74.6
	Men	79	25.4
Age	24 to 34	51	16.4
	34 to 44	97	31.2
	45 to 54	115	37
	55 and more	48	15.4
Teaching experience	4 years or less	38	12.2
	5 to 10 years	55	17.7
	11 to 20 years	104	33.4
	More than 20 years	114	36.7
Educational stage	Early Childhood and Primary	191	61.4
	Secondary and High School	120	38.6
Professional profile	Support teachers/Specialist (Therapeutic Pedagogy, Counselors, Hearing, and Speech)	47	15.1
	Other teachers (tutors, subject specialists, heads of study, principals, etc.)	264	84.9
Center by Funding/Ownership	Public	210	67.5
	Subsidized	101	32.5
Center by number of inhabitants in which is located	Equal or less than 3000	82	26.4
	From 3000 to 10,000 (included)	75	24.1
	From 10,000 to 100,000 (included)	63	20.3
	More than 100,000	91	29.3
Teaching experience with SEN students	4 years or less	116	37.3
	5 to 10 years	79	25.4
	11 to 20 years	81	26
	More than 20 years	35	11.3

**Table 4 children-09-00813-t004:** Kruskal-Wallis analysis by teachers’ age.

Variables	24–34 Years	35 to 44 Years	45 to 54 Years	More than 55 Years	*p*-Value
CV.6. The academic achievement of the SEN students is related to difficulties and barriers that must be overcome.	181.36	149.38	145.50	167.59	0.045
Cpa.3. Teachers feel empowered to facilitate the participation of families.	207.03	137.60	153.36	145.29	0.000
Cpa.4. There is flexibility in the class groups that encourages collaboration and communication between students.	188.47	152.06	146.40	152.47	0.026
Cpa.6. The center offers families various ways to motivate them to get involved in their children’s learning.	189.52	140.22	155.43	153.67	0.011
Cpa.7. The active participation of all students in the center’s life is facilitated.	183.16	145.25	151.25	160.25	0.049
Cpa.8. Families are informed of the progress of their children with SEN.	180.54	144.20	147.90	173.19	0.011
Total	192.46	148.57	146.19	155.78	0.011

Note. N between 24 and 34 years = 51; N between 35 and 44 years = 97; N between 45 and 54 years = 115; N more than 54 years = 48.

**Table 5 children-09-00813-t005:** Post hoc Kruskal-Wallis analysis according to the different age groups of teachers that are significant.

Variables	24–34 Years vs. 35–44 Years			24–34 Years vs. 24 to 54 Years			24–34 Years vs. +55 Years			35–44 Years vs. +55 Years			45–54 Years vs. +55 Years		
	*p*	24–34 r	35–44 r	*p*	24–34 r	45–54 r	*p*	24–34 r	+55 r	*p*	35–44 r	+55 r	*p*	45–54 r	+55 r
CV.6.	0.031	84.16	69.42	0.009	97.07	77.48									
Cpa.3.	0.000	95.76	63.32	0.000	104.11	74.36	0.001	59.16	40.27						
Cpa.4.	0.017	85.41	68.76	0.002	99.59	76.37	0.037	55.47	44.19						
Cpa.6.	0.002	88.84	66.96	0.013	96.67	77.66	0.022	56.00	43.63						
Cpa.7.	0.009	85.90	68.51	0.016	95.63	78.12									
Cpa.8.	0.007	85.58	68.68	0.010	95.73	78.08				0.037	68.58	81.94			
Total	0.006	87.50	67.66	0.001	101.44	75.54	0.040	55.52	44.14						

Note. N between 24 and 34 years = 51; N between 35 and 44 years = 97; N between 45 and 54 years = 115; N more than 54 years = 48.

**Table 6 children-09-00813-t006:** Kruskal-Wallis analysis according to teaching seniority.

Variables	<5 Years	5 to 10 Years	11 to 20 Years	>20 Years	*p*-Value
Cpa.3. Teachers feel empowered to facilitate the participation of families.	208.47	158.78	140.35	151.44	0.000
Cpe.2. Only the support specialists should attend to SEN students since they are the ones trained for it.	174.99	126.79	156.72	163.11	0.034
Cpe.4. All students can achieve the general objectives in an inclusive classroom with the necessary help.	178.09	180.94	142.86	148.59	0.016

Note. N less than 5 years = 38; N between 5 and 15 years = 55; N between 11 and 20 years = 104; N over 20 years = 114.

**Table 7 children-09-00813-t007:** Post hoc Kruskal-Wallis analysis according to the different age groups of teaching seniority.

Variables	<5 Years vs. 5–10 Years			<5 Years vs. 11–20 Years			<5 Years vs. >20 Years			5–10 Years vs. 11–20 Years			5–10 Years vs. >20 Years			11–20 Years vs. >20 Years		
	*p*	<5 r	5–10 r	*p*	<5 r	11–20 r	*p*	<5 r	>20 r	*p*	5–10 r	11–20 r	*p*	5–10 r	>20 r	*p*	11–20 r	>20 r
Cpa.3.	0.005	55.96	40.81	0.000	93.45	63.48	0.000	98.07	69.31									
Cpe.4.				0.032	83.24	67.21				0.009	92.52	73.38	0.021	96.97	79.22			

Note. N less than 5 years = 38; N between 5 and 10 years = 55; N between 11 and 20 years = 104; N over 20 years = 114.

**Table 8 children-09-00813-t008:** Analysis of the Mann Whitney U according to the educational stage of teaching.

	Early Childhood and Primary Education		Compulsory Secondary Education and Baccalaureate		Z	*p*-Value
	N	Rank	N	Rank		
CV.3. The work done by all students (with and without disabilities) is equally valued.	191	165.87	120	140.29	−2.559	0.010
CV.4. SEN students actively participate in all activities center (exhibitions, works, musical activities, theatrical performances …).	191	164.49	120	142.48	−2.333	0.020
Cpa.3. Teachers feel empowered to facilitate the participation of families.	191	167.81	120	137.20	−3.052	0.002
Cpa.4. There is flexibility in the class groups that encourages collaboration and communication between students.	191	165.56	120	140.79	−2.514	0.012
Cpa.5. The support provided by the Integration Support Classroom is coordinated with the work in the ordinary classroom.	191	170.00	120	133.71	−3.645	0.000
Cpa.6. The center offers families various ways to motivate them to get involved in their children’s learning.	191	164.23	120	142.90	−2.132	0.033
Total PartiCom	191	166.77	120	138.86	−2.756	0.006
Cpe.1. Teachers think that classrooms that have Sen students have lower academic levels than those that do not have these students.	191	147.14	120	170.10	−2.248	0.025
Cpe.2. Only the support specialists should attend to SEN students since they are the ones trained for it.	191	146.52	120	171.09	−2.406	0.016
Cpe.5. For attending SEN students, it is necessary to start from what they know how to do and not from the general level of the classroom.	191	163.62	120	143.87	−2.251	0.024

**Table 9 children-09-00813-t009:** Analysis of the Mann Whitney U according to the type of center’s funding/ownership.

	Public		Subsidized		Z	*p*-Value
	N	Rank	N	Rank		
CV.1. Attention to diversity is a priority in the educational project of the Center	210	149.13	101	170.28	−2.070	0.038
CV.2. Measures to address diversity are applied from inclusive policies.	210	142.66	101	183.74	−4.005	0.000
CV.5. SEN students are considered in both group interactions and tasks.	210	147.97	101	172.70	−2.480	0.013
CV.6. The academic achievement of the SEN students is related to difficulties and barriers that must be overcome.	210	143.33	101	182.35	−3.862	0.000
CV.7. Among the colleagues, there is a climate of collaboration and help to SEN students.	210	148.32	101	171.97	−2.347	0.019
CV.8. In the center, it is considered that the SEN students have the right to have education adapted to these needs.	210	147.58	101	173.50	−2.667	0.008
Cpa.1. Collaboration and coordination of teachers of different levels and subjects allow adequate attention to SEN students.	210	142.96	101	183.11	−3.888	0.000
Cpa.2. Teachers encourage collaboration and acceptance of all students in the classroom and center.	210	140.73	101	187.74	−4.738	0.000
Cpa.3. Teachers feel empowered to facilitate the participation of families.	210	144.11	101	180.72	−3.510	0.000
Cpa.4. There is flexibility in the class groups that encourages collaboration and communication between students.	210	143.07	101	182.89	−3.888	0.000
Cpa.5. The support provided by the Integration Support Classroom is coordinated with the work in the ordinary classroom.	210	148.93	101	170.70	−2.103	0.035
Cpa.6. The center offers families various ways to motivate them to get involved in their children’s learning	210	148.28	101	172.05	−2.287	0.022
Cpa.7. The active participation of all students in the center’s life is facilitated.	210	147.44	101	173.80	−2.672	0.008
Cpe.4. All students can achieve the general objectives in an inclusive classroom with the necessary help.	210	142.87	101	183.30	−3.872	0.000

**Table 10 children-09-00813-t010:** Analysis of the Mann Whitney U according to teachers’ professional profile.

	SEN Specialist		Others		Z	*p*-Value
	N	Rank	N	Rank		
CV.1. Attention to diversity is a priority in the educational project of the Center	47	132.49	264	160.19	−2.074	0.038
CV.5. SEN students are considered in both group interactions and tasks.	47	130.90	264	160.47	−2.267	0.023
CV.6. The academic achievement of the SEN students is related to difficulties and barriers that must be overcome.	47	129.47	264	160.72	−2.366	0.018
CV.7. Among the colleagues, there is a climate of collaboration and help to SEN students.	47	127.18	264	161.13	−2.577	0.010
CV.8. In the center, it is considered that the SEN students have the right to have education adapted to these needs.	47	123.12	264	161.85	−3.050	0.002
Total “*Inclusive values*”.	47	122.47	264	161.97	−2.916	0.004
Cpa.2. Teachers encourage collaboration and acceptance of all students in the classroom and center.	47	118.60	264	162.66	−3.396	0.001
Cpa.3. Teachers feel empowered to facilitate the participation of families.	47	111.40	264	163.94	−3.853	0.000
Cpa.4. There is flexibility in the class groups that encourages collaboration and communication between students.	47	117.36	264	162.88	−3.399	0.001
Cpa.6. The center offers families various ways to motivate them to get involved in their children’s learning	47	128.05	264	160.98	−2.422	0.015
Total “*degree of participation in the educational community.*”	47	129.39	264	160.74	−2.278	0.023

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
