# Peer review of "Inclusive Culture in Compulsory Education Centers: Values, Participation and Teachers’ Perceptions"

_children, 2022, doi:10.3390/children9060813_

Round 1

Reviewer 1 Report

This is an interesting manuscript on a relevant topic. I have identified some aspects that should be corrected or addressed.

In the Abstract (page 1, line 18, the authors wrote:  “We found significant differences were found in... “ (a reformulation is needed to avoid the repetition of the word “found“). Two dots are written at the end of the sentence.

Page 1, line 21: Two semicolons are written after “special educational needs (SEN);;”

Page 1, line 26: Consider changing… ”It’s essential elements are...” in “Its essential elements are... ”

Page 4, line 180: consider changing ”...the questionnaire is sent both online and on paper...” in ”the questionnaire was sent both online and on paper...”.

The purpose of the study should be mentioned at the end of the Introduction (no separate section is needed for this).

I think the Materials and Methods section would benefit from a more throughout description of the three categories announced in Table 1: inclusive values, participation of the educational community, and perceptions.

Authors should make sure that all the tables and figures are referred to in the text.

It is not clear what the colors in Figures 1, 2, and 3 represent. The authors should use a clear legend.

The limitations of the study should be discussed.

Author Response

Please see the attachment, and thank you for your suggestions and corrections.

Reviewer 2 Report

Important study as children with special needs have to be taken care by the society in a well-organized way. I am glad to assist with reviewing this article. 

Title is my opinion could be re-structured a little bit to give broader sense of the study. Abstract and key words are fine. 

Introduction is well-written, however, one thing that needs to be handled in the Introduction is the clarity of terms used in the study design. For example in their paper Special educational needs and disability Sue Keil and Olga Miller and Rory Cobb indicate the "confusion over the use of terms that represent differing ideological perspectives. Despite the social focus that characterises much of the discourse about disability, disability is frequently regarded as an aspect of special educational needs, an area in which a medical model is often dominant". I would suggest looking into the text. 

In Methods the description of the Participants selection is unclear. How was actually the sample group selected. The authors write about heterogeneous group (line 137 The data was triangulated to generate a series of data that was as heterogeneous and representative as possible in terms of characteristics, location, and typology of the educational Center" but how was it done in reality? and later "the centers and teachers were randomly selected according to the criteria determined in the independent variables, the willingness to collaborate freely and based on the census provided by the Provincial Directorate of Education of Leon (Spain)" - This needs to be described in more detail. 

There is nothing wrong with using self-made research tools especially that the authors provided some validity and reliability procedure. 

Results are described well, but in my opinions figures should be bigger so the reader can read the data on the diagrams more easily. Also Legend (1, 2, 3, 4, 5) needs to be defined as for the moment it is hard to ques what those numbers on the figures mean. 

I also have doubts whether you can use Mann-Whitney test for compering mean rank values? Even if you run this test for comparison should you put mean values of the variable in the tables (tab.8) not the rank values?   

 In Discussion a problem of accessibility should be tackled as well you discuss geographical location of the center as one of the factors, which is ok, but in this way your study is relevant only to local population while  different cultures will have different educational systems that treat the matter differently, which is reflected in pupils' chance for inclusion (look for example into: Examining the Extent to Which Learners with Special Educational Needs are Included in Regular Schools: The Case of Four Primary Schools in Cape Town, South Africa. 

And when you discuss SEN policy and SEN school population you narrow it to your local conditions. Maybe it would be worth comparing it to the situation elsewhere? Look for example into:" Construction of difference and diversity within policy and practice in England, Cambridge Journal of Education" or Teachers' Perceptions of Collaboration and Partnership regarding Children with Special Educational Needs in a Mexican Bilingual Elementary School, Global Studies of Childhood, and/or look into some gender specific issues in some countries (cultures) Gender bias and imbalance: girls in US special education programmes, Gender and Education'

There is also very narrow scope of references which are one-country originated, which is also an issue - you need to broaden the context of you findings and provide more international references.   

Author Response

(The authors gave the same response as above.)

Reviewer 3 Report

The authors note at the outset of their paper that many studies have been conducted into teacher's attitudes to the inclusion of pupils with special educational needs (lines 85-86).  But they provide little indication as to what their investigation will add to this topic. At best it will be a replication of previous findings from one region in Spain but this endeavour is deeply flawed for the following reasons.  Also the authors suggest that the purpose is “to identify the barriers that hinder inclusive education” (line 119) but I saw little evidence that this was achieved (see my comments below about the discussion group).

The quantitative aspect of the study uses a rating scale that has been developed by the authors and piloted with an unknown number of respondents (line 164).  However no details are provided of the psychometric properties of the scale nor was any confirmatory factor analysis undertaken with the much larger sample recruited to the present study.  In particular, test-retest statistics are not reported nor is the percentage of variance reported for the three factors and the item loadings.  No comparisons of the findings from this study can be made or imputed with previous research when very different measures are used.

In any event the authors report in unnecessary detail, item level statistics broken down by the chosen independent variables (Figures 1-3 and Tables 4-10). This is unnecessary because as their factor analyses suggests, many of the item ratings are correlated with one another.  Rather summary scores on each factor would suffice to explore possible relationships.  It is well known that item ratings tend to be more unreliable than summary scores.

The authors also ignore the inter-relationships among the independent variables although they do allude to the issue in the discussion (line 542-544).  To control for this, it is important that multi-variate statistics such as Multiple Regression are used but they failed to do this.  Had they used this approach, the paper would be considerably reduced in length and readers would get a greater appreciation of the variables of influence on teachers’ attitudes.  Even so, the sample size is a limiting factor (n=311)  and given the number of independent variables and sub-categories within them, their present sample is possibly under-powered.    The authors would be well advised to obtain the advice of an experienced statistician when designing studies like these and more especially when it comes to analysing the data.

The authors present little evidence as to the representativeness of the sample of teachers they obtained.  For example what are the teacher characteristics across the chosen schools – the total number; broken down by types of schools, gender etc. and how do these compare with the volunteer sample they obtained.  It is likely that their sample is biased in unknown ways.  Moreover they acknowledge that the chosen region may have different school types compared to other Spanish regions (lines 560 ff).

The authors report data obtained from a focus group of six ‘experts’ who seem to be involved in special education provision and are not therefore reflective of the bulk of respondents to the questionnaire.   If this truly was a mixed methods design, the focus group participants would have been ‘representative’ of those participating in the questionnaire phase.  So the barriers they report are not necessarily those of mainstream teachers for example.

Author Response

Please see the attachment and thank you for your revision, comments, suggestions, and corrections.

Round 2

Reviewer 2 Report

I can see that You have accommodated most of the comments and suggestions. Your paper is better now.

Reviewer 3 Report

See below.